# Socioeconomic Disparities in Hypertension by Levels of Green Space Availability: A Cross-Sectional Study in Philadelphia, PA

**DOI:** 10.3390/ijerph19042037

**Published:** 2022-02-11

**Authors:** Celina Koh, Michelle C. Kondo, Heather Rollins, Usama Bilal

**Affiliations:** 1Department of Epidemiology and Biostatistics, Drexel University Dornsife School of Public Health, 3215 Market St., Philadelphia, PA 19104, USA; ck876@dragons.drexel.edu; 2Urban Health Collaborative, Drexel University Dornsife School of Public Health, 3600 Market St., Philadelphia, PA 19104, USA; her43@drexel.edu; 3Northern Research Station, United States Department of Agriculture–Forest Service, 100 N. St., Ste 205, Philadelphia, PA 19103, USA; michelle.c.kondo@usda.gov

**Keywords:** green spaces, hypertension, socioeconomic status, health equity, Philadelphia

## Abstract

Green spaces have been proposed as equigenic factors, potentially mitigating health disparities. We used data from the 3887 participants residing in Philadelphia who participated in the Public Health Management Corporation’s Southeastern Pennsylvania Household Health Survey in 2014–2015 to assess whether socioeconomic disparities in hypertension are modified by availability of neighborhood-level green spaces. Socioeconomic status (SES) was measured using individual-level education and neighborhood-level median household income. Green space availability was measured using surrounding percent tree canopy cover, mean normalized difference vegetation index (NDVI), and proximity to nearest park. Using logistic regression models adjusted for age, sex, and race/ethnicity, we found that adults with higher educational attainment had significantly lower levels of hypertension (OR = 0.63, 0.57, and 0.36 for high school, some college, and college graduates, respectively, as compared to those with less than high school education), and this pattern was similar for median household income (higher prevalence in lower income areas). We found no significant interaction between education and percent tree canopy cover (*p* = 0.83), meaning that educational disparities in hypertension were similar across all levels of green space availability. These results held when using mean NDVI or distance to nearest park as availability measures, or when considering neighborhood-level median household income as the socioeconomic measure, although the specific patterns and significance of interactions varied by exposure and modifier. While socioeconomic disparities in hypertension are strong for adults residing in Philadelphia, green spaces did not seem to modify them.

## 1. Introduction

Green spaces can provide a wide variety of environmental, health, and equity benefits. Environmental benefits of green space, defined as land that is partially or completely covered in vegetation (e.g., grass, trees, shrubs, etc.) [1], include negating urban heat [2], minimizing air pollution [3], and regulating flooding [4]. Green spaces may also have health promoting benefits for outcomes such as mental health [4,5], physical activity and obesity [6,7,8], social cohesion [9], stress [10,11], restoration [12,13,14], pregnancy outcomes [15,16], and cardiovascular health [17,18,19,20,21,22,23,24,25]. Studies exploring the relationship between green spaces and health have used various measures for green space availability, including tree canopy cover, measures of green reflectance such as the Normalized Difference Vegetation Index (NDVI) or the Soil-adjusted Vegetation Index (SAVI), distance to various green spaces (e.g., parks), and presence of residential/neighborhood green spaces [26]. Since these measures represent different aspects of green space and its availability, and have exhibited differential associations with health outcomes, exploring multiple measures is key to better understand the linkages between green spaces and population health [27,28].

Another benefit of green spaces is a phenomenon known as the equigenesis (or equigenic) hypothesis of green spaces [29]. According to this hypothesis, green spaces promote health equity by supporting the health of residents in less advantaged neighborhoods and thereby reducing socioeconomic status (SES)-based disparities in cardiovascular mortality and mental health outcomes [4,5,30,31]. This mitigation of disparities is especially important as ensuring health equity is a cross-cutting theme for public health [32]. However, the measurement of SES is complex, as it constitutes a multidimensional construct, usually measured with indicators of income or wealth, education, and occupation [33]. 

There are wide socioeconomic disparities in cardiovascular disease (CVD) and its risk factors. For example, both the incidence and prevalence of CVD and its risk factors tend to be higher in people with lower income, lower educational attainment, and in working class individuals [34,35,36,37,38,39]. Specifically, people living in poverty, with less than a high school education, and who work in manual occupations tend to have a higher prevalence of hypertension [40]. Nevertheless, hypertension remains the leading risk factor driving preventable deaths globally [41]. In this study, we focus on hypertension disparities by educational attainment (the highest level of education an individual has completed) [42] and median household income [43] to further understand factors that may narrow these disparities. Investigating the potential relationship between SES-based disparities and hypertension prevalence could improve health outcomes by providing opportunities to reduce inequalities through evidence-based interventions.

A few studies have examined the influence of green spaces on the relationship between socioeconomic factors and CVD and its antecedents, including hypertension [25,29,30,44,45,46,47]. Overall, these studies found mixed results, with some of them showing clear equigenic effects of green spaces [29], some finding a widening of inequalities in greener areas [46,47], and some finding inconclusive results [25,30,44]. 

In order to further explore the equigenic effect of green spaces (narrower disparities in greener areas), and in view of conflicting results in previous research, we analyzed the relationship between green space availability and socioeconomic disparities in hypertension prevalence in Philadelphia from the period of 2014–2015. We leveraged data on multiple SES and green space indicators in an urban area with wide socioeconomic disparities, but with a committed effort to greening [48] and its equity consequences [49,50]. Based on previous studies and the potential effect of wide socioeconomic disparities across accessibility of green spaces and hypertension prevalence, we aimed to address two main hypotheses: (1) adults who have lower educational attainment will have greater odds of hypertension compared to those who have higher educational attainment and (2) adults who have higher green space availability will have narrower income-based and educational disparities in hypertension compared to those with lower green space availability (equigenic hypothesis).

## 2. Materials and Methods

### 2.1. Study Setting

We conducted a multilevel study of individuals nested in census tracts. The United States Census Bureau defines census tracts as statistical subdivisions of a county or equivalent [51]. We selected census tracts as the geographical unit of analysis due to accessibility of geocoding to complete, available, individual-level data. We obtained individual-level self-reported data on hypertension, age, sex, race/ethnicity, and education status from the Public Health Management Corporation’s Southeastern Pennsylvania Household Health Survey (SEPAHHS) 2014–2015 wave dataset. SEPAHHS includes information collected from adults (≥18 years old) living in Philadelphia, Bucks, Chester, Delaware, and Montgomery Counties via random digit dialing (landline and cell phone) administered in both English and Spanish [52]. The survey is conducted every 2 to 3 years and collects self-reported health status, health behaviors, and access to care [52]. The survey is designed to be representative of the adult non-institutionalized population of the area (see Table A4 for a comparison of the sample with the population of Philadelphia). Our study sample has a similar hypertension, gender, and race/ethnic distribution to the population of Philadelphia, with a higher proportion of individuals aged 50–64.

The final analytic sample included 3887 adults (18+ years old) that had valid responses on all relevant variables residing in 377 census tracts in Philadelphia County (Figure 1). Philadelphia is the poorest large US city and has wide health disparities [53,54,55,56]. Furthermore, we decided to restrict the sample to Philadelphia to narrow the focus to an urban area, as green space availability (and its management) differs between the surrounding suburban and exurban counties. Moreover, Philadelphia has had several greening efforts over the last few decades [48] and has considered how equity affects greening strategies [49] in the management of urban parks and green space [50], making this a policy-relevant study location. 

### 2.2. Outcome (Hypertension)

We defined hypertension status as self-reported high blood pressure, measured via responses to the survey question “Have you ever been told by a doctor or other health professional that you have high blood pressure or hypertension?” The available response categories were for yes, no, and only during pregnancy. We recategorized hypertension as a binary variable, operationalizing any observations that had reported hypertension only during pregnancy as no hypertension.

### 2.3. Exposure

We used two measures of SES: individual-level educational attainment and census tract-level median household income. Educational attainment was self-reported and obtained via survey response. Responses were categorized into four categories: less than high school, high school graduate (including high school graduates and General educational Development Test (GED)), some college or equivalent (including some college and technical, trade, and vocational school), and college graduate or higher. We selected these categories to be consistent with previous studies [30,34] and to avoid unbalanced categories with very low sample sizes. Median household income is another standard measure of SES. Individual-level income data were incomplete—missing for 869 out of the 3887 participants—which could cause potential issues if income was not missing at random. We therefore obtained census tract median household income from the 2012–2016 American Community Survey [57]. We were unable to account for 21 responses out of the 3887 participants due to missing median household income data per census tract (Figure A1). To make our analyses of this variable comparable with education (four categories), we operationalized median household income into quartiles (Table A1).

### 2.4. Effect Modifier (Green Spaces)

We used three measures of green space availability at the census tract level, based on previous research [20,23,30,34]: (1) percent tree canopy cover, (2) mean normalized difference vegetation index (NDVI), and (3) proximity to parks. First, we calculated tree canopy cover using raster-based tree canopy data derived from satellite imagery. The Chesapeake Bay Conservancy published high-resolution (1 m^2^) land cover data, including a tree canopy class, for the years 2014–2015 [58]. We computed percent tree canopy cover by dividing the tree cover area by the total area of each census tract. Second, we computed census tract-level mean NDVI using 250 m resolution data from the United States Geological Survey (USGS) for 2014–2015 utilizing the scale of −1 to 1 [59]. We recoded all values equal to −1 to No Data to remove water from our mean value calculations. Third, we calculated proximity to parks by computing the Euclidean distance from the centroid of each census tract of residence to the nearest park, where parks included community parks, farms, gardens, greenhouses, nurseries, linear parks, parkways, metropolitan parks, mini parks, neighborhood parks, regional/watershed parks, square/plaza parks, and watershed/conservation parks. We obtained location information for parks from the City of Philadelphia’s Park and Recreation Department Parks Assets dataset from 2019 [60]. The dataset includes city-owned buildings and facilities (i.e., community parks, mini parks, watershed/conservation park, etc.) [60]. Table A2 contains more details on our inclusion criteria. We conducted all spatial analyses and calculations using ArcGISPro version 2.9.1 [61]. We operationalized all green space indicators by categorizing continuous variables into tertiles (Table A1).

### 2.5. Covariates

We included age, sex, and race/ethnicity a priori as confounders based on a literature review and a directed acyclic graph. We included race/ethnicity as a proxy for racism, a key factor that affects hypertension, socioeconomic opportunities, and neighborhood accessibility to green spaces through residential segregation [62,63]. Race/ethnicity was categorized into 4 groups: non-Hispanic white (NHW), non-Hispanic Black (NHB), Hispanic/Latino (H), and Other. The “other” category, while collapsing widely distinct groups with different exposures, was needed since sample sizes for Asian (N = 62), Biracial/Multiracial (N = 139), and Native American (N = 24) were low. Age and sex were included as they are important risk factors for hypertension. Age was categorized into four groups (18–34, 35–49, 50–64, 65+). Similarly, sex was coded as a binary variable (male, female).

### 2.6. Statistical Data Analysis

The main objectives of this study were (1) to assess socioeconomic disparities in hypertension prevalence, and (2) to explore whether green space availability modifies socioeconomic disparities in hypertension prevalence. We set up a multilevel dataset, where each observation was an individual sampled in the SEPAHHS dataset, nested in a census tract. The dataset included individual-level variables (hypertension, age, sex, race, and education) and contextual-level variables (percent tree canopy cover, mean NDVI, proximity to nearest park, and median household income).

We fitted logistic regression models with hypertension (yes/no) as the outcome, adjusted for age, sex, and race/ethnicity, accounting for the survey clustering by census tract. In model 1, we included education as the key exposure. We hypothesized that individuals with lower educational attainment would have higher odds of hypertension. In model 2, we further adjusted for tertiles of tree canopy cover, and added an interaction between education and tree canopy cover. We hypothesized that educational disparities in hypertension would be narrower in areas with higher tree canopy cover. We tested the joint significance of interaction terms using the F test (categorical interaction in Figure A2). Furthermore, we also tested the significance of interactions using education or income as ordinal variables (ordinal interaction in Figure A2). We ran the following three secondary analyses: (1) changing the green space measure to mean NDVI, (2) changing the green space measure to distance to nearest park, and (3) changing the socioeconomic indicator to median household income.

To address the lack of independence between individuals living in the same census tract, all analyses used robust clustered standard errors at the census tract level. We also weighted all models to account for the balanced sampling weights of the SEPAHHS survey. All statistical analyses were conducted using R version 4.0.1 [64].

## 3. Results

The descriptive characteristics of the final analytic sample of 3887 adults (18+ years old) residing in Philadelphia from the years 2014–2015 are displayed in Table 1. Overall, the proportion of individuals with hypertension in the final analytic sample was 43.2% (N = 1679). Higher educational attainment was associated with an increased prevalence of hypertension, while it did not vary by levels of tree canopy cover, mean NDVI, and distance to nearest park. However, we observed large differences across age, race/ethnicity, and education in the prevalence of hypertension. Hypertension prevalence was highest among participants who were older, non-white, and had a lower educational attainment. 

Table 2 shows the results of model 1. We found a dose–response association between education and the odds of hypertension. Specifically, we found that, compared to individuals with less than high school education, after adjusting for age, sex, and race/ethnicity, individuals with high school education, some college, and college education had a 37% (95% CI: 10–56%), 43% (95% CI: 15–62%), and 64% (95% CI: 47–76%) lower odds of hypertension. Table A3 shows results for median household income, showing analogous findings. 

Figure 2 shows the results of model 2, estimating the association between hypertension and education by levels of tree canopy cover, adjusted for age, sex, and race/ethnicity. We found that education disparities in hypertension are similar across all levels of tree canopy cover (*p*-value for the joint test of all interaction coefficients = 0.772; *p*-value for the joint test of all interactions using ordinal education = 0.529). 

Lastly, we conducted two sets of sensitivity analyses (see Figure A2). First, we examined whether the patterns for education held for area-level median household income, an alternative SES indicator. While we found no significant interaction (*p*-value for the joint test of all interaction coefficients = 0.192; *p*-value for the joint test of all interactions using ordinal education = 0.459), we found no income-based disparities in areas with high tree canopy cover. Second, we examined the robustness of the analysis by exchanging the green space availability measurement from percent tree canopy cover to mean NDVI or distance to nearest park. For mean NDVI, results were also qualitatively similar to the main analysis (*p*-value for the joint test of all interaction coefficients = 0.961; *p*-value for the joint test of all interactions using ordinal education = 0.864). For distance to nearest park, we found a similar pattern to our main analysis, but with a significant interaction (*p*-value for the joint test of all interaction coefficients = 0.017; *p*-value for the joint test of all interactions using ordinal education = 0.068), driven by a much lower prevalence of hypertension in people with college education in medium green space availability areas.

## 4. Discussion

In this analysis of 3887 adults residing in Philadelphia, we found a very wide socioeconomic disparity in hypertension after adjusting for age, sex, and race/ethnicity. However, we did not find support for the equigenic hypothesis, as we found that the educational disparity in hypertension was similar across all levels of green space availability. While these results were robust to the choice of green space measure or socioeconomic indicator, we did observe some differences in results across these alternative specifications.

Our finding of socioeconomic disparities of hypertension aligns with findings from previous studies of a strong association between hypertension and measures of socioeconomic status (SES), including education and income [2,3,4,46,47]. However, we did not find support for the equigenic hypothesis of green spaces for our outcome of self-reported hypertension. Specifically, we found similar educational disparities in hypertension across all levels of tree canopy cover and proximity to parks. Our findings were concurrent with one study that found no effect modification for education for both systolic and diastolic blood pressure [25]. 

On the other hand, four previous studies have found support for the equigenic hypothesis [29,44,46,47]. These studies used other SES indicators including household income, individual income, occupational class, neighborhood deprivation, and neighborhood median household income, for which we found more support for the equigenic hypothesis. These studies also used a diversity of green space availability measures, including green space coverage, mean NDVI, and the number of local areas used for parks and/or recreational facilities. While our findings were qualitatively similar across measures of green spaces, we did find more support for the equigenic hypothesis when examining distance to nearest park as the green space measure. Two studies suggested that reduction in hypertension risk among lower SES populations is related to higher levels of green space, specifically mean NDVI [46,47]. Meanwhile, the other two studies indicate similar iterations of the equigenic hypothesis in other health outcomes such as all-cause mortality [29] and cardiovascular-related diseases [44]. Overall, our results may not necessarily support or refute the hypothesis, but they do provide an additional opportunity to explore this relationship further.

The main strength of this study is the multilevel nature of the design, with individual-level hypertension and education and neighborhood-level median household income, along with other confounders, and neighborhood-level measures of green space availability. Furthermore, we were able to test the robustness of our results using a different socioeconomic indicator (median household income) and measures of green space availability (mean NDVI and distance to nearest park). Lastly, all three models controlled for socioeconomic and demographic features while accounting for clustering within neighborhoods.

We also acknowledge several limitations. First, this is a cross-sectional study, and we were therefore unable to assess the temporality of associations or draw any causal inference. Second, hypertension was self-reported, which may introduce bias due to differential access to healthcare [65]. Third, we excluded 153 individuals because of missing data (4.2% of the eligible sample) and were precluded from using individual-level reported income as a marker of SES given higher levels of missingness. However, we were able to leverage neighborhood-level median household income to conduct a secondary analysis. Fourth, the generalizability of this study is limited to adults living in large urban areas, given our restriction of the study sample to adults residing in Philadelphia. Fifth, green space availability was measured at the census tract level rather than at the residential address level, which could lead to misclassification of availability, and thereby lead to biases in our estimates. Additionally, the green space data from the Chesapeake Bay Conservancy and the City of Philadelphia’s Parks and Recreation may be outdated in Philadelphia as urban green space design has been evolving towards renovated vacant lots and green roofs, which may not be included in either dataset [66]. Furthermore, the parks dataset does not include private green areas such as domestic gardens and backyards. We anticipate that the lack of inclusion of private green spaces introduces a possible issue of under-estimation of the effect of green space among affluent neighborhoods/participants. However, our analysis using cumulative green space (measured through mean NDVI) as the green space indicator showed similar results. Finally, we acknowledge that precent tree canopy, mean NDVI, and distance to nearest park do not fully represent green space availability as physical interaction/usage, accessibility, and urban integration should be considered as well.

## 5. Conclusions

The present study contributes to the current body of evidence by providing additional insight on the association between the proximity to green space and educational disparities in hypertension prevalence. Green space is a relatively novel tool for addressing health equity and outcomes for population health, and future studies can continue to improve on the measurement of both green spaces, markers of health disparities, and other health outcomes of interest.

## Figures and Tables

**Figure 1 ijerph-19-02037-f001:**
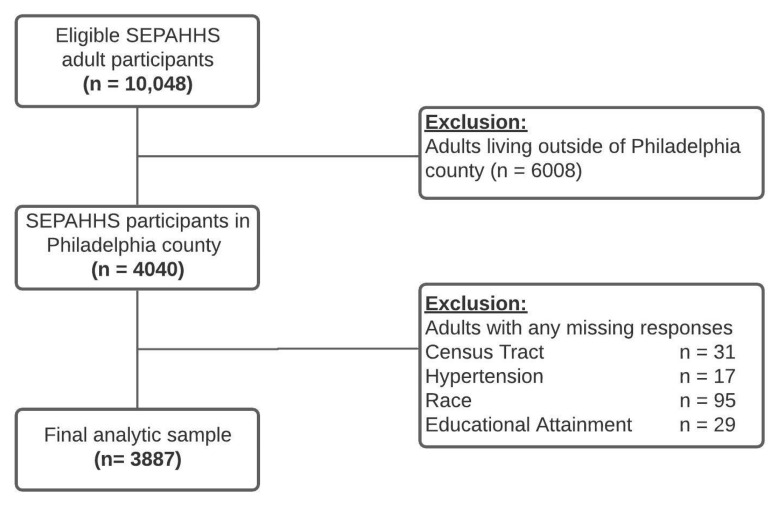
Exclusion criteria of the final analytic sample.

**Figure 2 ijerph-19-02037-f002:**
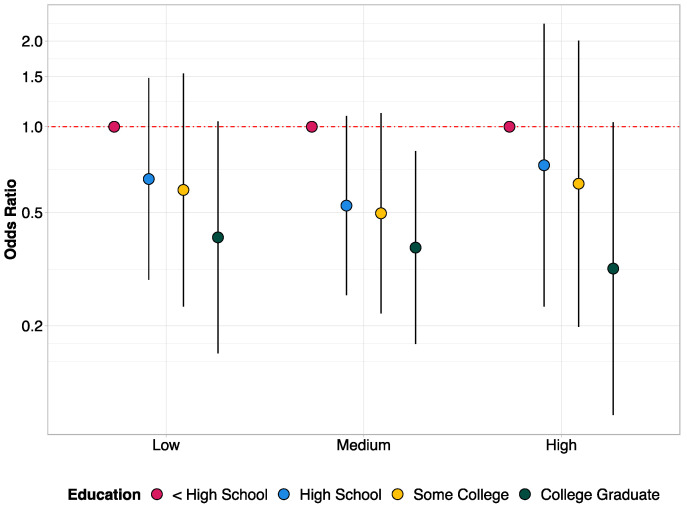
Comparisons of education disparities in hypertension across tertiles of tree canopy cover (Model 2). Note: Odds ratios are represented as the interaction terms between tree canopy cover and education. The reference group is less than high school education.

**Table 1 ijerph-19-02037-t001:** Descriptive characteristics of included study participants.

	Hypertension		
	No (N = 2208)	Yes (N = 1679)	Overall (N = 3887)
Tree Canopy Cover			
Low	682 (57%)	514 (43%)	1196
Medium	701 (53%)	617 (47%)	1318
High	825 (60%)	548 (40%)	1373
Mean NDVI			
Low	686 (61%)	439 (39%)	1125
Medium	668 (50%)	679 (50%)	1347
High	854 (60%)	561 (40%)	1415
Distance to Nearest Park			
Low	700 (57%)	532 (43%)	1232
Medium	774 (57%)	578 (43%)	1352
High	734 (56%)	569 (44%)	1303
Age (years)			
18–34	462 (87%)	67 (13%)	529
35–49	771 (75%)	263 (25%)	1034
50–64	667 (48%)	716 (52%)	1383
65+	308 (33%)	633 (67%)	941
Sex			
Male	797 (58%)	586 (42%)	1383
Female	1411 (56%)	1093 (44%)	2504
Race/Ethnicity			
NH White ^1^	1087 (65%)	598 (35%)	1685
NH Black ^1^	757 (46%)	874 (54%)	1631
Hispanic/Latino	213 (61%)	135 (39%)	348
Other	151 (68%)	72 (32%)	223
Educational Attainment			
Less than high school	142 (37%)	240 (63%)	382
High school graduate	670 (51%)	654 (49%)	1324
Some college or equivalent	531 (57%)	406 (43%)	937
College graduate or higher	865 (70%)	379 (30%)	1244
Median Household Income			
Q1	361 (46%)	416 (54%)	777
Q2	599 (54%)	517 (46%)	1116
Q3	627 (59%)	440 (41%)	1067
Q4	609 (67%)	297 (33%)	906
Missing	12 (57%)	9 (43%)	21

Note: All percent values displayed are row precents. This table shows raw counts and proportions (unweighted). ^1^ NH indicates non-Hispanic.

**Table 2 ijerph-19-02037-t002:** Association between hypertension and education adjusted for age, sex, and race/ethnicity (Model 1).

	OR (95% CI)	*p*-Value
Educational Attainment		
Less than high school	1.00 (Ref.)	
High school graduate	0.63 (0.44; 0.90)	0.012 *
Some college or equivalent	0.57 (0.38; 0.85)	0.005 *
College graduate or higher	0.36 (0.24; 0.53)	<0.001 *
Age (years)		
18–34	1.00 (Ref.)	
35–49	3.07 (2.02; 4.66)	<0.001 *
50–64	10.92 (7.57; 15.75)	<0.001 *
65+	20.38 (13.82; 30.04)	<0.001 *
Sex		
Male	1.00 (Ref.)	
Female	0.96 (0.78; 1.19)	0.73
Race/Ethnicity		
NH White ^1^	1.00 (Ref.)	
NH Black ^1^	2.28 (1.79; 2.89)	<0.001 *
Hispanic/Latino	1.17 (0.77; 1.76)	0.465
Other	0.85 (0.53; 1.37)	0.513

Note: Coefficients are shown as odds ratio (OR), and 95% confidence interval (CI) from a weighted logistic regression model using generalized estimating equation (GEE). ^1^ NH indicates non-Hispanic. * *p* < 0.05.

## Data Availability

Data provided by The Public Health Management Corporation’s Community Health Database. In any event, this publication does not imply any representation that Public Health Management Corporation endorses or supports any position or views of Drexel University Dornsife School of Public Health. The Pennsylvania Land Cover of the Chesapeake Bay Conservancy and the Parks & Recreation Assets datasets are publicly available.

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
