# Peer review of "Socioeconomic Disparities in Hypertension by Levels of Green Space Availability: A Cross-Sectional Study in Philadelphia, PA"

_ijerph, 2022, doi:10.3390/ijerph19042037_

Round 1
Reviewer 1 Report
This paper analyzed the relationship between green space exposure and educational disparities in hypertension prevalence in Philadelphia from 2014-2015. Overall, the topic is interesting and important to the public and the manuscript is well prepared. I am only concerned about the indicators of urban green space. I would suggest the authors try another indicator of green space, i.e., NDVI. The NDVI is a commonly used green exposure assessment indicator in assessing the relationship between green space and public health. In addition, the authors can obtain the NDVI data that matches with the study period of self-reported data. This can avoid the limitations as stated in 216-225.
Reviewer 2 Report
This is an interesting and succinctly written study of the relationship between green space, hypertension, and educational level in Philadelphia. This is an important area of study, and one which I feel will be worth publishing once it has been improved. In particular, there are several major limitations to the reporting and methods which need to be addressed. My recommendations for improvement are outlined as follows:
- It does not seem that the study measures ‘exposure’, but rather ‘availability’, as stated in the title. The title therefore seems more representative than the terms used in the manuscript. I would suggest the term ‘exposure’ be amended to ‘availability’ throughout, as I would argue that measuring distance to green space is measuring ‘availability’ rather than ‘exposure’. This study does not measure whether people use the spaces, but how much is available to them (which is fine, it's just important to be clear on this in the presentation)
- All key terms should be defined in the introduction. This mostly relates to ‘green space’
- Please ensure ‘green space’ is spelled consistently throughout- it appears as ‘greenspace’ in some places
- The introduction is very brief and should be expanded to discuss why the authors believe the associations they are investigating might exist. Why hypertension? Why education?
- Because of the brief introduction, the hypotheses seem to be very sudden and not thoroughly justified
- Why was the study restricted to Philadelphia?
- Why were these four educational categories chosen? For example, higher degrees were not included
- Exposure: The section on income is well-justified but should be moved to the covariates section
- Line 96: the list of States isn’t really needed
- Line 99: how was proximity measured? Euclidean or network distance? To the boundary, centroid, or entry point of each park? Was only one park included, and why? Why was the size of the parks not included?
- Please provide full citations for all software used
- Why were tree canopy and distance converted to tertiles?
- Covariate: please justify more fully why these covariates were chosen
- Why were no economic indicators included? You make a good case for omitting income, but employment or local area economics (area-averaged income, deprivation) would make suitable alternatives. Economics have been shown to confound/interact with associations between green space and health, so it is important that this is addressed. I would strongly recommend adding in at least one of these measures to the analysis.
- Please begin your Results section by fully describing the sample
- How do these hypertension and educational statistics compare to the general population?
- Tables 2: please indicate which variables were statistically significant in the model
- You should consider generalisability in the limitations section
Round 2
Reviewer 2 Report
Thank you to the authors for responding to most of the reviewers’ comments, and the quick turnaround on the paper. There are, however, a couple of points which have not been adequately addressed, which are mostly just adding a little more justification and explanation into the introduction and methods:
- In the introduction (line 64), you state that your work is ‘in view of conflicting’ previous studies- please expand how and why your study would be able to overcome these limitations
- Please introduce the idea of educational attainment in the introduction, and explain why you hypothesise that this might be related to green space
- You include multiple measures of green space, but green space is only introduced as a singular concept in the introduction. Please explore the different measures, and the reasons for this, before they are added as effect modifiers. Why is it important to measure green space in these different ways? What insights to they provide? Any why is this relevant to hypertension? These are all ideas which should be explored in the introduction, which is still quite brief.
- Following on from the above, on Line 167: why did you hypothesise that educational disparities in hypertension would be narrower in areas with higher tree canopy cover?
- Results: please fully describe your sample and compare the gender/age/hypertension balance to that of the general population
